# On the loss of context-awareness in general instruction fine-tuning

## Abstract

Pretrained Large Language Models (LLMs) require post-training methods such as supervised fine-tuning (SFT) on instruction-response pairs to enable instruction following. However, this process can potentially harm existing capabilities learned during pretraining. In this paper, we investigate the loss of context awareness after SFT, defined as the capability to extract and understand information from the user-provided context and respond accordingly. We are the first to identify and show that the loss of context-awareness appears on instruction-finetuned LLMs when the chat template is applied to the input prompts. We identify the performance decline is partially caused by the bias embedded into the chat template to focus less on the user-provided context. Based on these observations, we propose two methods to mitigate the loss of context awareness in instruct models: post-hoc attention steering on user tokens and conditional instruction fine-tuning with a context-dependency indicator. Empirical experiments on 4 context-dependent downstream tasks and 3 pretrained LLMs of different sizes show that our methods effectively mitigates the loss of context awareness without compromising the general ability to follow instructions. Our findings also strongly advocate the necessity to carefully benchmark context awareness after instruction fine-tuning.

## 1 Introduction

Large Language Models (LLMs) pretrained on large-scale datasets acquire a diverse set of language modeling capabilities in pretraining. To enhance these models' ability to follow general instructions, further fine-tuning is typically required such as supervised instruction fine-tuning (SFT) (Wei et al., 2021; Ouyang et al., 2022) and reinforcement learning from human feedback (RLHF) (Christiano et al., 2017) to better understand and respond to human requests. However, the additional fine-tuning can potentially harm existing capabilities learned in pretraining, as pointed out by several existing works (Lin et al., 2024; Bai et al., 2022; Fu et al., 2024).

In this paper, we particularly investigate the loss of context-awareness after instruction fine-tuning, which is the capability to understand and retrieve exact information from the user-provided context and respond accordingly. Context awareness is crucial for many real-world use cases, including retrieval augmented generalization (RAG) (Khandelwal et al., 2020; Izacard et al., 2023; Xu et al., 2023b), in-context learning (Agarwal et al., 2024), and contextual question-answering (QA) (Rajpurkar et al., 2016; Choi et al., 2018; Dua et al., 2019). We first illustrate the loss of context awareness in Figure 2 with the Needle-in-a-Haystack test on four popular instruction-tuned models. We demonstrate that the performance degradation is consistent on both long-context and relatively short-context LLMs, which cannot be solely explained by the distribution difference in context lengths between the instruction dataset and the evaluation benchmarks, as suggested by prior works (Dubey et al., 2024).

We identify that the bias embedded within the chat template to focus less on the user tokens is a major cause of context-awareness degradation. Normal instruction fine-tuning dataset contains a mixture of both model-dependent and context-dependent queries. The former consists of queries the model can respond to relying mostly on its **internal knowledge** learned during pretraining. On the other hand, responding to the second type of query requires exact information retrieval and processing from the **user-provided context** in the input prompt, such as in-context learning, long-form instruction-following, and contextual QA tasks. However, a query accompanied by a context

can still be a model-dependent query as the model may have learned the context during pretraining, and is able to respond without relying on the user given context. Therefore, it is challenging for the model to differentiate between these two type of queries from the prompt only, and incorrect identification could lead to hallucination or being over-reliant on user-provided context

We validate the bias embedded in chat templates with the Needle-In-a-Haystack test (NIH) (Kamradt, 2023), which requires a model to retrieve a given text "needle" from a long paragraph of irrelevant text. Our experiments show that the needle retrieval performance drops on instruction-finetuned models *only* when the chat template is applied to the input, which is however crucial for the model to distinguish different roles in conversations. We further show that the performance drop can be attributed to the drop in attention value allocated to the whole user input section. Therefore, the context retrieval capability is not lost in the model, but "inhibited" by the chat-formatted fine-tuning.

Based on these observations, a straightforward approach is to directly steer the attention value during inference time to emphasize the user inputs on instruction-tuned models. This can be achieved by manually intervening attention scores of the user context tokens during generation. Experiments show that while performance on simple retrieval tasks can be significantly boosted by attention steering, manipulating the attention value in the inference stage can harm other capabilities of the model, deteriorating performance on more complex tasks.

To further improve upon the undesirable trade-off of post-hoc techniques, we are motivated to steer the attention allocation in the fine-tuning stage. To achieve this goal, we identify context-dependent conversations before fine-tuning and add a special token to the prompt as a hint to the model. The special token can then be added at inference time when more attention is demanded to be allocated to the user-provided context. Empirical experiments show the effectiveness of our method on several open-source, pretrained LLMs and instruction fine-tuning datasets.

Our contributions are summarized as follows:

- We identify that supervised instruction fine-tuning causes pretrained language models to deteriorate in context awareness (even for short context lengths).
- We pinpoint the worsened context awareness is associated with attention allocation bias embedded within the chat template.
- We propose an inference-time technique to partially recover the context-awareness of general instruction-tuned language models by manually intervening attention scores.
- We propose a training-time technique utilizing conditional indicators to mitigate the loss of context awareness of pretrained language models during instruction-tuning.

## 2 RELATED WORK

**Instruction fine-tuning and chat templates.** Large language models only learn language modeling on general corpus during pretraining. To enable instruction-following, they usually require supervised fine-tuning on instruction-following datasets (SFT) (Wei et al., 2021; Ouyang et al., 2022), followed by reinforcement learning with human feedback (RLHF) (Christiano et al., 2017). In this paper, we mainly focus on the SFT stage. Instruction fine-tuning datasets consist of user instruction and target model response pairs, which can be collected from modified NLP tasks (Wei et al., 2021; Longpre et al., 2023), human annotations (Ouyang et al., 2022; Chiang et al., 2023) or synthesized data from existing LLMs (Ding et al., 2023; Xu et al., 2023a). Instruction fine-tuning usually converts training examples into a dialog format with a chat template, which consists of a user role indicator (e.g. `[INST]` in llama-2 models), an assistant role indicator (e.g. `[/INST]` in llama-2 models) and an optional system role indicator (e.g. `<<SYS>>` in llama-2 models). Other role indicators are also used in more complicated scenarios such as tool using (Schick et al., 2023). However, these role indicators and role partition in the conversation are not sufficiently presented and learned during pretraining, making them prone to bias during fine-tuning.

**Evaluation and improvement of context-awareness** The capability to retrieve and understand information from the context and respond accordingly is important for many tasks including in-context learning (Agarwal et al., 2024; Brown et al., 2020), retrieval augmented generation (Lewis

et al., 2020) and contextual QA (Dua et al., 2019; Rajpurkar et al., 2016; Choi et al., 2018). These tasks are commonly included in standard evaluation benchmarks of recent LLMs. To improve the performances on context-dependent tasks, a common practice is to collect or synthesize context-dependent data, especially contextual QA data, and mix them into the fine-tuning dataset (An et al., 2024; Dubey et al., 2024). Apart from adding more data, Hsieh et al. (2024) also explores calibrating the attention weights to compensate for the drop of attention weight in the middle of a long context. However, to date, limited existing works have examined how chat templates affect context-awareness when fine-tuning language models on conversational instruction-following data. The most relevant conclusion from existing works is mentioned by Dubey et al. (2024) that long-context capabilities learned during pretraining drops significantly after SFT. However, they only attribute this performance decline to a lack of long-context data in SFT stage.

**Conditional Supervised Fine-Tuning**   Conditional Supervised Fine-Tuning (CSFT) involves fine-tuning a pretrained model on specific tasks while conditioning on additional information or context. The conditions represent specific attributes or styles of the demonstration response and are concatenated to the original input as prefixes. The model is expected to associate the condition prefix with the style or attribute such that whenever the condition prefix is added, the model generates responses accordingly. Dong et al. (2023) proposes SteerLM, where they applied CSFT to align models with human values (e.g. helpfulness, humor, and creativity) by conditioning the model on attribute prefixes. Korbak et al. (2023) adds two control tokens as indicators for good and bad demonstration responses. As opposed to attribute tags or control tokens, Chain of Hindsight (Liu et al., 2023) conditions demonstration responses on natural language comments on the quality or styling of the response.

## 3   LOSS OF CONTEXT-AWARENESS EMBEDDED IN CHAT TEMPLATES

### 3.1   LOSS OF CONTEXT-AWARENESS IN NEEDLE-IN-A-HAYSTACK (NIH) TEST

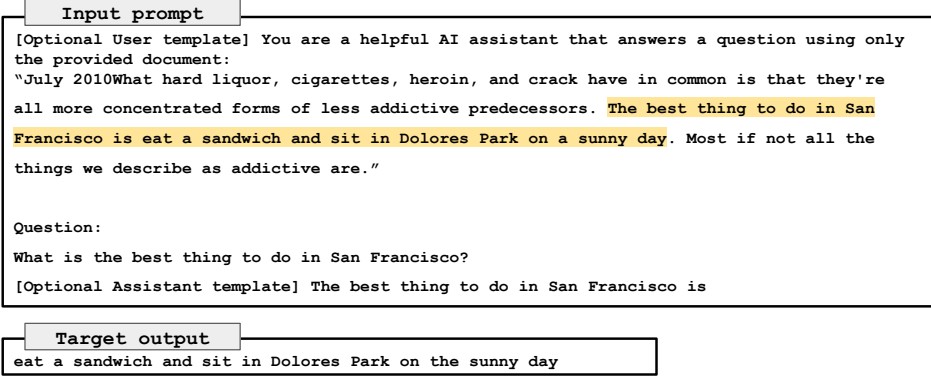

Figure 1: An example of the Needle-in-a-haystack (NIH) test used in our work. [Optional User template] and [Optional Assistant template] are user and assistant role indicators used in instruction finetuned models. The inserted needle is highlighted in yellow.

In this section, we show the loss of context awareness after instruction fine-tuning with the needle-in-a-haystack (NIH) test.

**Experimental settings.**   NIH evaluates the performance of language models to extract a given sentence (the needle) from an irrelevant context. The needle can be inserted at different locations in contexts with different lengths. We report the recall error:

$$\text{err} = \frac{1}{|K|} \sum_{w \in K} \mathbb{1}(w \in output)$$

where $K$ is the set of keywords in the targeted output and $output$ is the output of the LLM. We average the recall error across 400 NIH tests with different insertion locations and context lengths within the model's context window. More details about the NIH tests can be found in Appendix A.2.

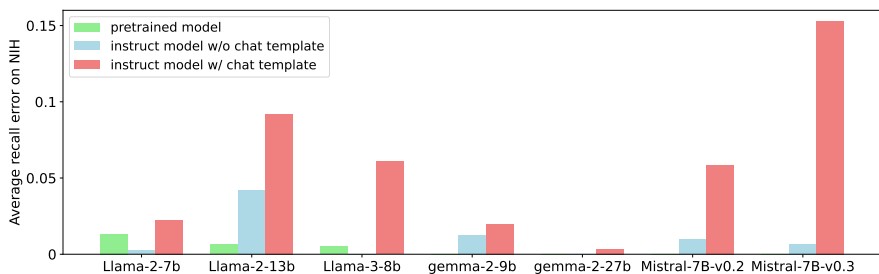

Figure 2: Average recall error (1 - recall) on NIH for different model series. We report the performance of official instruct models (with and without chat templates) and their corresponding pretrained models from 5 model families with sizes ranging from 7B to 27B. Recall errors for some models are too small to be visible from the figure. Detailed numerical values can be found in Appendix B.1

We evaluate the results on four open-source pretrained language models (not instruction-tuned) and their corresponding instruction-finetuned versions from Llama and Mistral series. Here we don't consider stronger close-source models as their pretrained versions are not available. The context window lengths of these models range from 4,096 to 32,768. We show a typical NIH prompt used in our experiments in Figure 1. When the chat template is applied to the prompt, the whole input prompt is partitioned into the user instruction input and model response, indicated by special role markers in the chat template (e.g. `<|user|>` and `<|assistant|>`). These chat templates are learned only in the instruction fine-tuning data, to teach the model to perform dialog conversation in response to the user instructions.

**Results.** We report the results in Figure 2. As we can see from the figure, the recall error increases significantly on instruction finetuned models when the chat template is added. When the chat template is removed, the recall error on the instruction-tuned model is comparable to or even better than pretrained models. These results indicate that the context retrieval capabilities are not wiped away in the model during instruction fine-tuning, but instead impacted by the bias embedded in the chat templates. However, simply removing the chat templates is not practical as they are necessary for the model to distinguish different roles in conversations. Therefore, we are motivated to recover the context-awareness of instruction-finetuned models by mitigating the negative impacts of chat templates on context-dependent tasks.

Moreover, the aforementioned phenomenon is consistent among models with different context window lengths, different model families and chat templates, and model sizes. In the remaining sections, we will focus on the context awareness of relatively short-context LLMs.

## 3.2 ATTENTION ALLOCATION BIAS FROM CHAT TEMPLATES

Based on the observation that the performance on NIH only drops when the chat template is present, we hypothesize that the chat templates, which are newly introduced in instruction fine-tuning, may encode bias to downweight the importance of user inputs when generating responses.

The distribution of the dataset adopted for pretraining and instruction-tuning is fundamentally different. In the pretraining stage, the model is trained on general-domain texts where all tokens are equally important in the context. However, during instruction fine-tuning, the model learns to distinguish between user input and its responses through chat templates. This user and model partition is not predominantly present in the pretraining corpus, and therefore difficult for the model to benefit from pretraining to learn a generalizable mapping. As a result, the model may tend to memorize the global bias to attend less on the user input if the instruction finetuning dataset is dominated by queries that do not require exact information retrieval.

To validate our hypothesis above, we visualize the attention allocated to each part of the model input in the NIH test, with and without the chat template. An input to an instruction-finetuned LLM

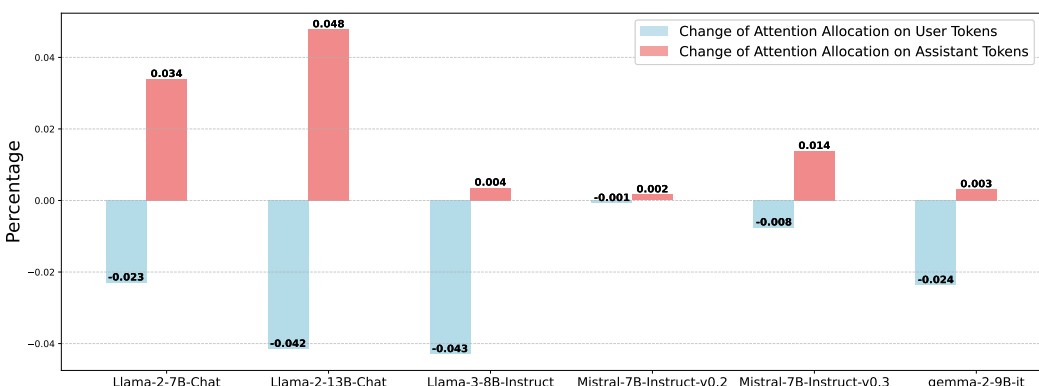

Figure 3: We visualize the changes in attention allocation on user tokens and assistant tokens after applying the chat templates. The attention allocation is calculated when the model is generating the first answer token in its response. For the case where the chat template is applied, we normalize the attention value on user tokens, assistant tokens, and the BOS token such that attention scores allocated to the three sums up to 1. The attention weight is averaged on 400 tests with context lengths ranging from 200 to 4000 and needle depth from 0% to 100%. Details for absolute attention scores allocated to each parts can be found in Appendix B.2.

can be divided into four parts: chat template, including role indicators used to separate user inputs and model response; special tokens, specifically the BOS token denoting the beginning of the text; user input and model response prefix. We calculate and visualize the changes of attention weight allocation on 6 models from 5 model families in Figure 3. Specifically, we calculate the attention weight from the last token before the answer token, which denotes the word "is" in Figure 1, to other tokens in the input prefix. We calculate the attention weight allocation on the most representative head for context retrieval, which is the head with the highest attention weight on the user input. From the figure, we can see a clear trend that the attention weight allocated to user tokens decreases when the chat template is added and attention allocated to assistant tokens increases.

The decreased attention weight allocated to user input context implies the model is relying more on its knowledge when generating the next token. For queries that require information retrieval from the user-provided context, the probability of hallucination also increases.

## 4 MITIGATING THE LOSS OF CONTEXT-AWARENESS

In the previous sections, we have established that the loss of context awareness to the user input is apparent when the chat template is applied and is directly reflected in the attention allocated. The next step is to mitigate this loss. The first straightforward approach is to directly intervene and increase the attention weight allocated to the user input to compensate for the attention allocation impacted by the chat template. We call this attention steering in this paper, which can be generally performed at inference time, regardless of how the model is fine-tuned. However, since the attention layers encode additional information beyond merely context dependency, manipulating the attention weights (especially with a larger magnitude) may also introduce unintended artifacts.

Therefore, in the second method, we want to encourage the model to focus more on the user input during training for context-dependent queries. Recall from Section 1 that both model-dependent and context-dependent queries can have context provided in the input, the model is unable to reliably distinguish between these two types of queries from the input prompts only. Therefore, we propose a metric to identify context-dependent queries with a seed LLM. We then explicitly inform the model of *when* to attend more to the user-provided context by appending an indicator token to the selected subset of context-dependent instructions during fine-tuning.

## 4.1 Post-hoc Attention Steering on User Input

In this section, we explain in detail how to increase the attention allocation level on the user input with post hoc modification to the attention weight during model inference.

Specifically, on each transformer layer, we modify the self-attention weight:

$$\hat{\text{Att}}(\mathbf{x}, \mathbf{y}) = \begin{cases} \frac{1}{Z} \cdot \text{Att}(\mathbf{x}, \mathbf{y}) & \text{if } \mathbf{y} \in U \\ \frac{1}{Z} \cdot \alpha \text{Att}(\mathbf{x}, \mathbf{y}) & \text{otherwise,} \end{cases} \tag{1}$$

where $\mathbf{x}$ and $\mathbf{y}$ are two tokens in the input sequence, $0 < \alpha < 1$, $U$ is the subset of user input tokens intended to emphasize, and $Z$ is the normalization constant.

We adopt the same implementation of attention steering as Zhang et al. (2024b). They steered the attention of pretrained language models to emphasize a user-specified portion of the user instruction, enabling models to follow user instructions without explicit instruction fine-tuning. In our setting, we up-weight the attention of instruction fine-tuned models on the whole user input prompt which consequently down-weights other tokens (chat template role tokens, BOS/EOS tokens, and partially generated model responses).

Although post hoc attention steering to improve context-awareness can be generally applied during inference time after a language model has been instruction fine-tuned, it negatively impacts other capabilities of the model that also depend on the attention score. This is a fundamental constraint of post-hoc model editing techniques, as components within a language model simultaneously provide multiple different functions. The trade-off of compromising the performance of other functions limits the magnitude of steering, which in turn limits the effectiveness of improving context awareness.

In the next section, we propose a training-stage method to train a model to better respond to context-dependent queries with an indicator token optionally provided by the user.

## 4.2 Instruction Fine-tuning with Context-dependency Indicators

Instead of post-hoc attention steering which introduces negative side effects, it is more desirable to directly control the attention during the fine-tuning process. Our goal is to nudge the model during training to teach the model to pay more attention to the user input when an indicator is appended to the user instruction by the user.

Specifically, we process the instruction fine-tuning examples such that responses that demand *more* attention on user instruction context are prepended with a special indicator token in its corresponding user instruction. Thus, after training whenever this indicator token is added and the model generation is conditioned on this token, the model will attend more to the user-provided context.

### 4.2.1 Identifying context-dependent Examples

We first introduce a metric to identify context-dependent training examples from a general instruction fine-tuning dataset. A training sample in the instruction fine-tuning dataset is a conversation between the user and model assistant, which may consist of multiple turns of instruction and response pairs. For a sample with $n$ turns of instructions and responses, a maximum of $n$ indicator tokens can be prepended in front of each of the responses to indicate this particular turn of response demands significant user instruction as context.

Let us denote user instructions as $X$, assistant responses as $Y$, and a conversation of $n$ total turns as $C = [X_1, Y_1, \ldots, X_n, Y_n]$. We start by preparing a seed instruction-finetuned model $M$, which can be the same or a weaker pretrained model fine-tuned with the original instruction fine-tuning dataset. We then define the context-dependency score for the $m^{\text{th}}$ turn response $Y_m$ as follows

$$s_M(Y_m) = \frac{1}{|Y_m|} \sum_{\mathbf{y} \in Y_m} \max_{h \in H} ( \sum_{\mathbf{x} \in X_1 \cup \ldots \cup X_m} \text{Att}_h(\mathbf{y}, \mathbf{x})), \tag{2}$$

where $H$ is the set of attention heads in model $M$. As different heads learn different capabilities, we keep the max user attention weight across all heads to select the most representative head for attention allocation on the user input. The final score measures the sum of attention scores allocated to all user instructions in prior turns $X_1 \cup \ldots \cup X_m$, averaged over response tokens $y \in Y_m$. In

| Datasets | Avg. conversation length | # conversations | # instructions | # instructions in $\hat{S}$ |
|---|---|---|---|---|
| ShareGPT | 1,567.68 | 93,645 | 331,722 | 38,542 |
| UltraChat-200k | 1,437.33 | 207,865 | 657,794 | 108,646 |
| WizardLM-70K | 484.00 | 57,523 | 57,523 | 12,938 |

Table 1: Statistics of instruction fine-tuning datasets in our experiments. # denotes the number of conversations and individual instructions in the dataset. We report the statistics after performing preprocessing as detailed in Section 5.1.2. Average length is measured in the number of tokens with TinyLlama tokenization. $\hat{S}$ is the selected subset of context-dependent conversation turns.

practice, we compute the score on a single middle layer for efficiency. We defer the discussion on layer selection to the Appendix B.3.

### 4.2.2 INSTRUCTION FINE-TUNING WITH CONTEXT-DEPENDENCY INDICATORS

With the context-dependency scores calculated for each model response in the instruction fine-tuning dataset, a threshold $0 < \beta < 1$ can then be selected such that each conversation turn $(\mathbf{X}_m, \mathbf{Y}_m)$ with $s_M(\mathbf{Y}_m) > \beta$ is considered highly context-dependent and added into the subset $\hat{S}$.

For each conversation turn $(\mathbf{X}_m, \mathbf{Y}_m) \in \hat{S}$, we append a special token [IND] to the user instruction $\mathbf{X}_m$. In our implementation, the special token [IND] is added as an additional special token to the vocabulary to avoid conflicting with existing ones. The modified subset $\hat{S}$ is then reincorporated into the instruction fine-tuning dataset as the final training data. Our method is more general in terms of the task and input format compared to related works, which synthesize context-relevant examples targeted toward specific tasks (e.g. contextual QA). This makes our method more applicable to different types of user queries that require more attention to the user-provided context.

## 5 EXPERIMENTS

We compare our method to vanilla instruction-tuning to demonstrate the efficacy of mitigating loss of context awareness. We instruction-finetuned 3 open-source, pretrained models on 3 different instruction datasets and evaluated the models on both context-dependency and general instruction-following tasks.

### 5.1 EXPERIMENTAL SETTINGS

#### 5.1.1 MODELS

We consider 3 open-source, pretrained large language models in our experiments: TinyLlama-1.1B (Zhang et al., 2024a), Llama-2-7B (Touvron et al., 2023), and Llama-3-8B (Dubey et al., 2024). TinyLlama-1.1B is a 1.1B Llama model pretrained on 3 trillion tokens with a context window length of 2048. Llama-2-7B and Llama-3-8B have a context window of 4096 and 8192 tokens, respectively. Due to limited computational resources, we only include Llama-2-7B and Llama-3-8B in our experiments and truncate the training examples up to 4096 tokens. We fine-tune Llama-2 and Llama-3 models using QLoRA (Dettmers et al., 2024) on transformer layers. Detailed hyperparameters can be found in Appendix A.1.1.

#### 5.1.2 INSTRUCTION FINE-TUNING DATASETS

We include 3 open-source instruction fine-tuning datasets: ShareGPT datasets adopted by Vicuna (Chiang et al., 2023), UltraChat-200k (Ding et al., 2023), and WizardLM-70K (Xu et al., 2023a). For ShareGPT, we follow the same preprocessing process as Chiang et al. (2023). We also removed refusal responses in ShareGPT and WizardLM-70K as the fine-tuned models will become oversensitive otherwise.

For all 3 datasets, we removed model responses without user input instructions in incomplete conversation chunks. Statistics of the processed datasets are presented in Table 1.

5.1.3 BENCHMARKS AND METRICS

| SFT Dataset | Model Name | $\alpha$ | NIH | SQuAD | QuAC | DROP |
|---|---|---|---|---|---|---|
| ShareGPT (Vicuna) | TinyLlama | 1.0 | 0.9846 | 0.5918 | 0.1130 | 0.2739 |
| | | 0.9 | **1.0** | **0.5972** | **0.1140** | **0.2781** |
| | | 0.8 | 0.4661 | 0.3657 | 0.0810 | 0.2539 |
| | Llama-2 | 1.0 | 0.3378 | 0.7601 | 0.1590 | **0.3390** |
| | | 0.9 | **0.4018** | **0.7920** | **0.1740** | 0.3309 |
| | | 0.8 | 0.2686 | 0.7699 | 0.1670 | 0.3353 |
| | Llama-3 | 1.0 | 0.8957 | 0.8216 | 0.1560 | **0.4215** |
| | | 0.9 | **0.91** | **0.8307** | **0.1650** | 0.4110 |
| | | 0.8 | 0.5975 | 0.5363 | 0.0650 | 0.2057 |
| UltraChat-200K | TinyLlama | 1.0 | **1.0** | **0.7475** | 0.1570 | **0.3096** |
| | | 0.9 | **1.0** | 0.7408 | **0.1590** | 0.3016 |
| | | 0.8 | 0.8368 | 0.6243 | 0.1400 | 0.2801 |
| | Llama-2 | 1.0 | **0.9850** | 0.8272 | 0.1540 | **0.3791** |
| | | 0.9 | 0.9429 | **0.8529** | **0.1700** | 0.3774 |
| | | 0.8 | 0.3303 | 0.7739 | 0.1180 | 0.2261 |
| | Llama-3 | 1.0 | **1.0** | 0.8393 | 0.1610 | **0.5099** |
| | | 0.9 | **1.0** | **0.8604** | **0.1720** | 0.4952 |
| | | 0.8 | 0.9182 | 0.7095 | 0.1450 | 0.2801 |
| WizardLM-70K | TinyLlama | 1.0 | 0.9250 | **0.5994** | 0.0990 | **0.2753** |
| | | 0.9 | **1.0** | 0.5878 | **0.1040** | 0.2697 |
| | | 0.8 | 0.6132 | 0.4624 | 0.0510 | 0.2484 |
| | Llama-2 | 1.0 | 0.7375 | 0.8229 | 0.1550 | **0.3407** |
| | | 0.9 | **0.8264** | **0.8430** | **0.1820** | 0.3360 |
| | | 0.8 | 0.5621 | 0.8144 | 0.1850 | 0.3326 |
| | Llama-3 | 1.0 | **0.9846** | 0.8765 | 0.1560 | **0.4687** |
| | | 0.9 | 0.9700 | **0.8851** | **0.1650** | 0.4203 |
| | | 0.8 | 0.9625 | 0.7173 | 0.1050 | 0.3042 |

Table 2: NIH and reading comprehension performances with different attention intervention factors.

**Benchmarks on context-awareness** In addition to NIH, we additionally include 3 closed-book QA tasks to benchmark context awareness: SQuAD (Rajpurkar et al., 2016), QuAC (Choi et al., 2018) and DROP (Dua et al., 2019).

SQuAD is a reading comprehension benchmark where the answer to each question can be found in the context. We only evaluate the answerable subset of questions in SQuAD 1.0. QuAC is similar to SQuAD but the questions are more open-ended and the answers contain a longer span in the context. We only evaluate the model on the first conversation round of QuAC, since WizardLM-70K is a single-round conversational dataset. DROP requires a more comprehensive understanding and analysis of the given context and the answers require retrieval of multiple information from the context and discrete operations such as addition, sorting, or counting. As instruction fine-tuned models are not specifically trained on QA tasks to provide concise answers, the models' responses are generally more verbose. Therefore, we report the containing score (whether the model response contains the ground-truth answer) instead of the F1 score to exclude the impacts of response styles of different models. Prompt templates for QA tasks are listed in Appendix A.3.

For Needle-in-a-haystack, we report the recall error as defined in Section 3.1. We set the maximum NIH context length to 1k for models fine-tuned on WizardLM-70K, due to the shorter instructions in the dataset. For models fine-tuned on ShareGPT and UltraChat-200k, we set the maximum NIH context length to the maximum context window considered in finetuning, which is 2k for TinyLlama and 4k for Llama-2 / Llama-3. The prompt template used in NIH is the same as Section 3.1 except that we remove the response prefix and keeps only the user input prompt.

**Benchmark on general instruction-following**   We evaluate the general instruction-following performance after instruction fine-tuning on MT-Bench (Zheng et al., 2023). The quality ratings of the responses are judged by a GPT-4 judge regarding the helpfulness, relevance, accuracy, depth, creativity, and level of detail of the response. We report the average rating across 80 responses.

## 5.2 ATTENTION STEERING ON USER INPUT

We report the performance of direct attention steering in Table 2. According to the results, a medium intervention factor $\alpha$ can boost the performance of NIH and most QA tasks except DROP. DROP is a more challenging closed-book QA dataset that requires additional operations or calculations based on information retrieved from the context. The worsening performance on DROP suggests that intervening attention scores, albeit improving context awareness, might hurt other capabilities of the model. For general instruction-following tasks that does not require heavy context-dependency, the user can choose to turn off the attention steering, and keep the same performance on general tasks.

## 5.3 INSTRUCTION FINE-TUNING WITH CONTEXT-DEPENDENCY INDICATORS

| SFT dataset | Pretrained Model | Method | NIH | SQuAD | QuAC | DROP | Avg. | MT-Bench |
|---|---|---|---|---|---|---|---|---|
| ShareGPT (Vicuna) | TinyLlama-1.1B | Vanilla | 0.9846 | 0.5918 | 0.1130 | 0.2739 | 0.4908 | 3.7250 |
| | | + Indicator | **0.9921** | **0.6144** | **0.1290** | **0.2784** | **0.5035** | **3.8250** |
| | Llama-2-7b | Vanilla | 0.3378 | 0.7601 | 0.1590 | **0.3390** | 0.3990 | **6.4875** |
| | | + Indicator | **0.7007** | **0.7830** | **0.1600** | 0.3390 | **0.4957** | 5.7375 |
| | Llama-3-8b | Vanilla | 0.8957 | 0.8216 | 0.1560 | 0.4215 | 0.5737 | **7.4375** |
| | | + Indicator | **0.9404** | **0.8394** | **0.1660** | **0.4317** | **0.5943** | 7.1875 |
| UltraChat-200K | TinyLlama-1.1B | Vanilla | **1.0** | 0.7289 | 0.1520 | 0.3096 | 0.5476 | 3.9000 |
| | | + Indicator | **1.0** | **0.7475** | **0.1570** | 0.3096 | **0.5535** | **4.0250** |
| | Llama-2-7b | Vanilla | **0.9850** | 0.8272 | 0.1540 | **0.3791** | 0.5863 | 5.7125 |
| | | + Indicator | 0.9725 | **0.8508** | **0.1570** | 0.3758 | **0.5890** | **5.8250** |
| | Llama-3-8b | Vanilla | **1.0** | 0.8393 | 0.1610 | **0.5099** | 0.6276 | 7.2375 |
| | | + Indicator | **1.0** | **0.8510** | **0.1660** | 0.5022 | **0.6298** | **7.2750** |
| WizardLM-70K | TinyLlama-1.1B | Vanilla | 0.9250 | 0.5994 | 0.0990 | 0.2753 | 0.4747 | 4.2750 |
| | | + Indicator | **0.9925** | **0.6279** | **0.1140** | **0.2836** | **0.5045** | **4.5625** |
| | Llama-2-7b | Vanilla | 0.7375 | 0.8229 | 0.1550 | 0.3407 | 0.5140 | 5.7750 |
| | | + Indicator | **0.9254** | **0.8260** | **0.1640** | **0.3444** | **0.5650** | **5.9875** |
| | Llama-3-8b | Vanilla | 0.9846 | 0.8765 | 0.1560 | 0.4687 | 0.6215 | 7.1125 |
| | | + Indicator | **0.9871** | **0.8792** | **0.1710** | **0.4785** | **0.6290** | **7.2750** |
| (Internal datasets) | Llama-2-7b-chat | - | 0.8264[*] | 0.8301 | 0.1330 | 0.4422 | 0.5579 | 6.9375 |
| | Llama-3-8b-Instruct | - | 1.0[*] | 0.8612 | 0.1850 | 0.4654 | 0.6279 | 8.0750 |

[*] Here NIH is evaluated without the response prefix used in Section 3.1 and the maximum context length is set to 4096 for fair comparison. Therefore, the numbers are different from the numbers in Figure 2.

Table 3: Comparing vanilla instruction finetuning with finetuning with context-relevant indicators (+ indicator). On '+ Indicator' models, `[IND]` is added for NIH and contextual QA tasks and removed for MT-Bench. As a reference, we also list the performances evaluated on official Llama-2 and Llama-3 instruct models, which are finetuned with internal datasets.

**Settings and selected subset $\hat{S}$.**   We adopt a TinyLlama model finetuned on the original ShareGPT (Vicuna) dataset as the seed model $M$. We compute the context-dependency score on an arbitrary middle layer (15 in all of our main experiments) for faster computation. As we show in Appendix B.3, subsets selected by scores calculated on different middle layers are highly consistent. We compute the context-dependency scores on all three instruction fine-tuning datasets and append `[IND]` to prompts associated with responses with context-dependency scores above the threshold $s_M(\mathbf{Y}_m) > \beta$. We set $\beta = 0.6$ in all our main experiments. Ablation studies on the threshold $\beta$ can be found in Appendix B.4. We report the numer of selected instructions of each fine-tuning dataset in Table 1. More statistics can be found in Appendix B.5.

**Evaluation results on context-awareness.**   We report the performance of NIH and three QA tasks in Table 3. For the "vanilla" fine-tuning setting, we train and evaluate the model without the in-

dicator token. For the "+ Indicator" setting, we add the indicator token to the selected subset of prompts in fine-tuning and all queries for evaluation. According to the table, "+ Indicator" outperforms "vanilla" fine-tuning in most cases. Particularly, "+ Indicator" consistently outperforms direct attention steering on DROP, reinforcing that the training-time solution handles trade-offs between language model capabilities better than inference-time editing. The results demonstrate that models can learn to focus more on the user-provided context when the indicator token is present in the prompt.

**Evaluation on general instruction-following benchmarks.** We also report the evaluation results on MT-Bench in Table 3. As samples in MT-Bench are mostly open-ended questions without heavy context dependency, the context-dependency indicator is not added for the "+ Indicator" models during evaluation. Models finetuned with our method show comparable or even better performance on MT-Bench in most cases. Therefore, our methods can effectively mitigate the loss of context awareness without losing the general ability of instruction following. The context indicator enables more fine-grained control over language models' behaviors during inference time.

## 6 CONCLUSION

This work highlights the detrimental effects of supervised instruction fine-tuning on the context-awareness of pretrained language models, even in scenarios involving short context lengths. We have identified that this decline is closely linked to inherent attention allocation biases within chat templates. To combat these challenges, we propose an inference-time technique that allows for the manual adjustment of attention scores, facilitating a partial recovery of context awareness in instruction-tuned models. Furthermore, we introduce a training-time approach that employs conditional indicators to help preserve context awareness during the instruction-tuning process. Together, these contributions aim to enhance the performance of language models in maintaining contextual understanding while benefiting from supervised instruction.

**Limitation** Our technique of associating context-dependent user instructions with the indicator token may also encode other unintended styles of the selected subset of instructions. The issue is particularly aggravated if the subset of context-relevant samples is small and significantly different from the remaining dataset. Utilizing the indicator token also requires the users to know which instructions require paying more attention to the user context during inference time. Tasks demonstrated in our experiment (e.g. keyword retrieval and closed-book QA) evidently require more attention to the user context. However, complex tasks may involve multiple language skills which complicates whether the addition of the indicator token will benefit the performance. A future direction is to automatically determine whether appending the indicator to a user instruction is necessary. Besides, we only validate the effectiveness of our proposed methods on relatively small-size models due to limited computational resources.

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

# A    APPENDIX

## A.1    EXPERIMENTAL DETAILS

### A.1.1    INSTRUCTION FINE-TUNING

We adopted the fine-tuning recipes from the Huggingface alignment-handbook[1] for Llama-2 and Llama-3 QLoRA tuning. For the TinyLlama model, we use the fine-tuning recipe provided by the author[2]. We finetune the models for 1 epoch on ShareGPT and UltraChat-200K and 2 epochs on WizardLM-70K as it has a smaller training set. We use the TinyLlama chat template for all instruct models finetuned in Table 2 and 3.

## A.2    NIH EVALUATION DETAILS

For all NIH evaluations, we average the recall error on 400 tests. Specifically, we evaluate on 20 context lengths uniformly distributed between 200 and the maximum context length, and 20 needle insertion depths uniformly located within 0% and 100%.

## A.3    CONTEXTUAL QA EVALUATION DETAILS

We list the prompts used in contextual QA tasks in Table 4 and Table 5. For contextual QA tasks, we generate answers up to 100 tokens and truncate them at the end of the first complete sentence. For NIH tests, we generate the answers up to 50 tokens.

As UltraChat-200K constructs their data with a fixed set of prompt templates similar to our default ones used in evaluation (The templates used for ShareGPT and WizardLM models in Table 5 and 4), we evaluate UltraChat-200K finetuned models with a simpler template to exclude the impact from overfitting on finetuning prompt templates.

| Instruct Finetuning Dataset | Template for SQuAD and DROP |
|---|---|
| ShareGPT & WizardLM-70K | {context}\nAnswer the question according to the above passage: {question} |
| UltraChat-200K | {context} {question} |

Table 4: Prompt templates used for SQuAD and DROP in Table 2 and Table 3 when the model is finetuned on different instruction finetuning datasets.

| Instruct Finetuning Dataset | Template for QuAC |
|---|---|
| ShareGPT & WizardLM-70K | {context}\nAnswer the question with pieces from the the above passage: {question} |
| UltraChat-200K | {context} {question} |

Table 5: Prompt templates used for QuAC in Table 2 and Table 3 when the model is finetuned on different instruction finetuning datasets.

# B    ADDITIONAL EXPERIMENT RESULTS

## B.1    FULL NIH RESULTS ON OPEN-SOURCE OFFICIAL MODELS

In Figure 2, we only report the NIH performances when the response prefix is added for fair comparison. In Table 6 we show the exact numbers for Figure 2 as well as additional evaluation results without the response prefix. When the response prefix is removed, the performance drop on NIH is even more significant compared to without chat templates.

---

[1]https://github.com/huggingface/alignment-handbook
[2]https://github.com/jzhang38/TinyLlama

| Model Name | Context window | w/o chat template w/ response prefix | w/ chat template w/ response prefix | w/o response prefix |
|---|---|---|---|---|
| Llama-2-7b Llama-2-7b-chat | 4096 | 98.67% 99.75% | - 97.75% | - 82.71% |
| Llama-2-13b Llama-2-13b-chat | 4096 | 99.35% 95.78% | - 90.79% | - 92.71% |
| Llama-3-8b Llama-3-8b-instruct | 8192 | 99.50% 100% | - 95.25% | - 95.35% |
| mistral-v0.2 mistral-v0.2-instruct | 32768 | 100% 99.00% | - 94.14% | - 93.92% |
| mistral-v0.3 mistral-v0.3-instruct | 32768 | 100% 99.32% | - 84.71% | - 72.00% |
| gemma-2-9b gemma-2-9b-it | 8196 | 100% 98.75% | - 98.03% | - 98.25% |
| gemma-2-27b gemma-2-27b-it | 8196 | 100% 100% | - 99.64% | - 99.25% |

Table 6: NIH performance with and without chat templates on different models.

## B.2 FULL RESULTS FOR FIGURE 2

In Figure 3, we only show the changes of attention allocation with and without chat templates. In Figure 4 we show the absolute numbers of attention allocation to each part of input prompts. When the chat template is added, we normalize the attention weight on user tokens, response tokens and BOS token only, with a sum of attention allocation to be 1.

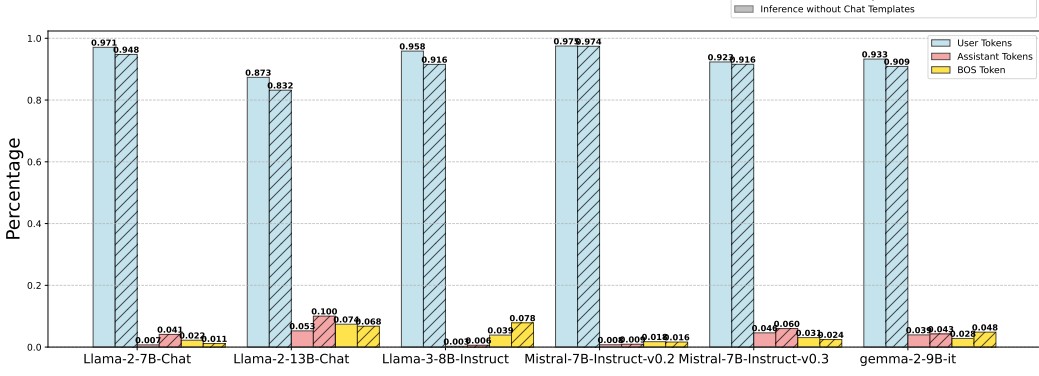

Figure 4: We visualize the full attention allocation on user tokens, assistant tokens and BOS token with and without applying the chat templates. The attention allocation is calculated when the model is generating the first answer token in its response. For the case where the chat template is applied, we normalize the attention value on user tokens, assistant tokens, and the BOS token such that attention scores allocated to the three sums up to 1. The attention weight is averaged on 400 tests with context lengths ranging from 200 to 4000 and needle depth from 0% to 100%.

## B.3 AGREEMENT BETWEEN DIFFERENT LAYERS

In Figure 5, we calculate and visualize the disagreement heatmap in $\hat{S}$ selection when the context-dependency score is calculated on different layers. We use the same TinyLlama model fine-tuned on the vanilla ShareGPT dataset as the seed model $M$. Specifically, we first calculate the context-dependency scores for each conversation turn in 500 randomly sampled examples from the ShareGPT dataset on different layers. We then select the top 10% conversation turn with the highest context-dependency scores on the layer as the subset $\hat{S}$. We compute the disagreement between two

Figure 5: We visualize the disagreement heatmap of $\hat{S}$ selection when the context-dependency score $S_M(\mathbf{Y}_m)$ is calculated on different layers. We select 10% of conversation turns with the highest context-dependency scores on each layer as $\hat{S}$. The disagreement is measured by the number of non-overlapped conversation turns in $\hat{S}$ selected by any two layers.

layers by calculating the ratio of non-overlapped conversation turns in their selected $\hat{S}$. We can see from the figure that the disagreement between 9 middle layers are low, indicating that we can safely choose an arbitrary layer for the context-dependency score calculation.

### B.4 ABLATION STUDY FOR DIFFERENT THRESHOLD $\beta$

| Threshold $\beta$ | SQuAD | QuAC | DROP | MT-Bench |
|---|---|---|---|---|
| 1.0 (Vanilla) | 0.5918 | 0.1130 | 0.2739 | 3.725 |
| 0.5 | 0.6207 | 0.1270 | 0.2872 | 4.075 |
| 0.6 | 0.6144 | 0.1290 | 0.2784 | 3.825 |
| 0.7 | 0.6160 | 0.1290 | 0.2786 | 3.675 |

Table 7: Ablation study with different threshold $\beta$, which is used in Section 4.2.

We use $\beta = 0.6$ in all our main experiments. To evaluate the sensitivity to the threshold $\beta$, we select $\hat{S}$ with different thresholds and prepare the final modified instruction finetuning dataset. We finetune a TinyLlama-1.1B model on these three datasets and evaluate them on three contextual QA

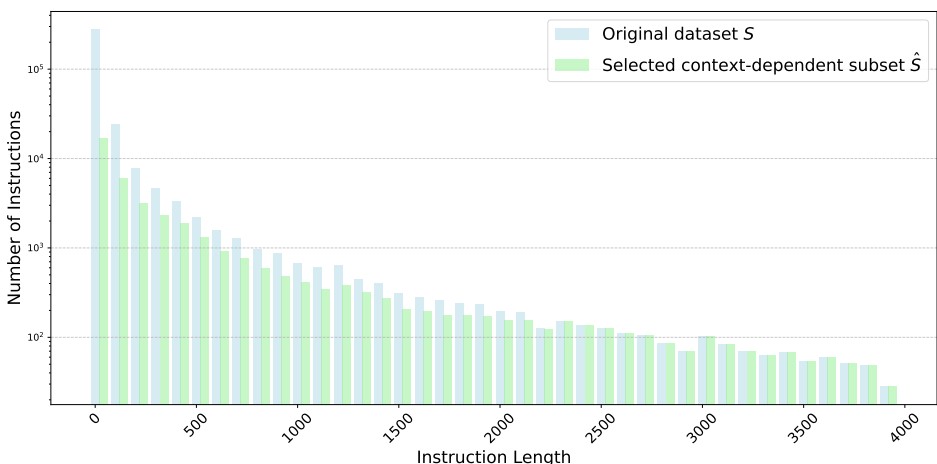

Figure 6: Change of instruction lengths between the original and the selected subset from ShareGPT dataset.

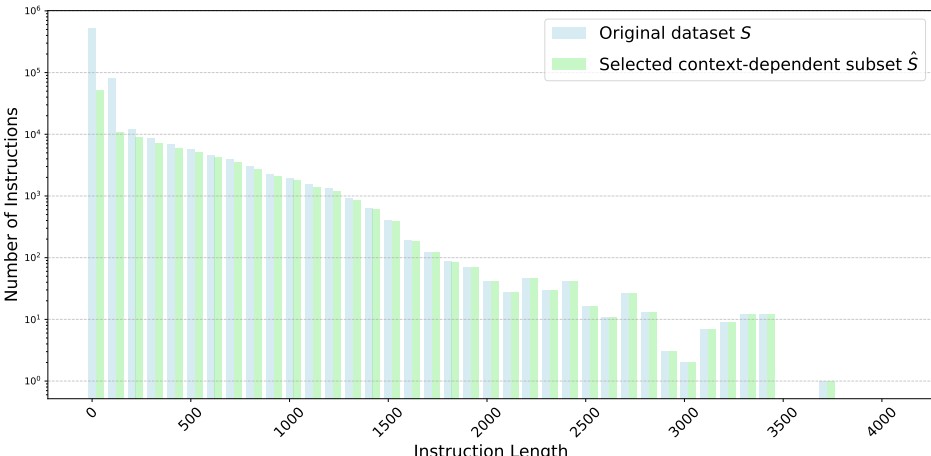

Figure 7: Change of instruction lengths between the original and the selected subset from UltraChat-200K dataset.

tasks and MT-Bench. As we can see from Table 7, all three models outperforms vanilla finetuning on the contextual QA tasks. However, performance on MT-Bench shows a decreasing trend when the threshold increases from 0.5 to 0.7, potentially due to a more drastic difference between $\hat{S}$ and the unselected subset.

## B.5 DISTRIBUTION OF INSTRUCTION LENGTHS

Here we visualize the change of distribution of instruction length between original instruction fine-tuning dataset and the selected context-dependent subset $\hat{S}$. Although a higher context-dependency is to some extent correlated with longer instruction length, there are still a large amount of short instructions showing high context dependency and selected into $\hat{S}$.

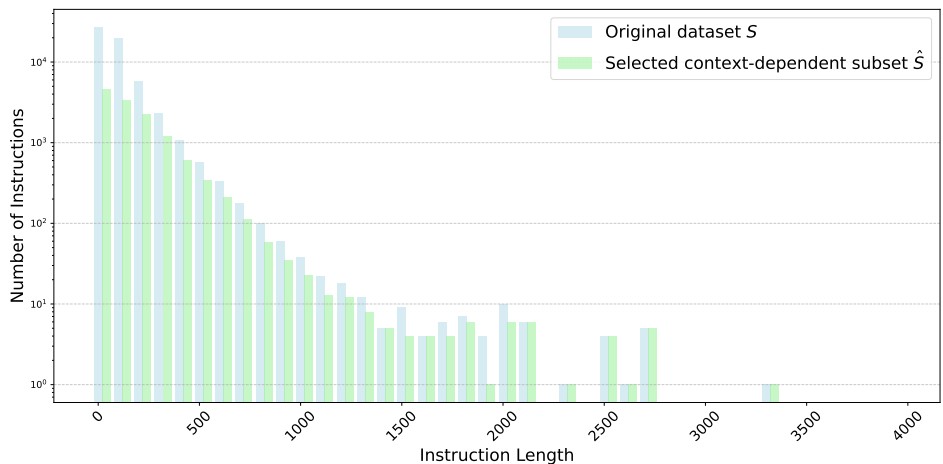

Figure 8: Change of instruction lengths between the original and the selected subset from WizardLM-70K dataset.

