# OpenReview forum: "On the loss of context-awareness in general instruction finetuning"
_ICLR.cc/2025/Conference — ICLR 2025 Conference Withdrawn Submission_

### Official Review · Reviewer_1kxK · 2024-10-24

**Soundness:** 2
**Presentation:** 2
**Contribution:** 1
**Rating:** 5
**Confidence:** 4

**Summary:**

This paper studies the problem of lower context awareness ability in instruction-tuned LLMs. The authors first conduct the needle-in-a-haystack (NIH) test on pre-trained and instruction-tuned LLMs to demonstrate the problem and then attribute this issue to the overemphasis on chatting templates by analyzing the distributions of attention activations. They further propose two strategies to enhance the ability to understand context during runtime or training time.

**Strengths:**

1. This paper conducts experiments on various LLMs from diverse model families.
2. Exploring the weaknesses of instruction-tuned LLMs is good.

**Weaknesses:**

1. The motivation of this work is unclear. As the goal of SFT is to achieve a _helpful_ chatbot, why do we have to care about their poorer context awareness? For example, in Table 3, we can find that a model with better context-awareness performance (i.e., NIH/SQuAD, QuAC, DROP) doesn't necessarily lead to a better instruction-following ability (i.e., MT-Bench). Also, in the NIH example of Figure 1, even though an instruction-tuned LLM cannot provide exactly the same suggestion from the user inputs at the first run, it doesn't necessarily mean that this instruction-tuned LLM lost its ability to retrieve knowledge from the context because it could be the instruction-tuned LLM fail to follow your first instruction that "answer question based on the given paragraph", or even could be the model feels that the target retrieved sentence is not helpful enough. A more comprehensive study on this phenomenon, at least, should provide diverse prompts that emphasize the idea of "answer questions based on the given paragraph" in different levels, and the instruction-tuned LLM _constantly_ fails to retrieve the target sentence.

2. The claim that poor context awareness can be attributed to the chat template is not supported. Firstly, attention weights may not faithfully express LLMs behaviors [1], especially in the cases where Transformers has skip-connect architecture. Secondly, even though I accept attention weight as a tool, the authors didn't set up a baseline to define when an attention weight is "high" and when it should be considered "low." Since we have formatted the user prompts with templates (meaning that the template is used), some attention weights have been allocated to the template part, which is reasonable. So, at least, we need to define a quantifier to measure the significant levels of attention weights. Specifically on Figure 3 (left), I feel the orange bar (User part) on Raw is almost equivalent high to that of Templated. Thirdly, the authors only cherry-pick one self-attention head from a model to conclude their findings, which is not reliable. Finally, instruction-tuned models pay some attention to the chat template, which is intuitive as they are trained on the data with the template; I cannot see any logical connection between the templates and the context awareness (I mean, I cannot prove/derive any connection between these two phenomenons, so they could be correlated, but may not be causally).

3. The authors didn't report the general instruction-following performance (i.e., MT-Bench) with the attention intervention strategies. So, we are unsure whether this strategy will hurt the generalizability of instruction-tuned models.

4. Some of the improvements shown in Table 3 are not significant. For example, QuAC of UltraChat on Llama-2 (0.154 --> 0.157), SQuAD of WizardLM on Llama-2 (0.8229 -> 0.8260) on Llama-3 (0.8765 -> 0.8792), and so on.

Overall, I am not satisfied with the proposed methods' motivations, main findings, and effectiveness of the proposed methods.

[1] Jain, Sarthak, and Byron C. Wallace. "Attention is not Explanation." Proceedings of the 2019 Conference of the North American Chapter of the Association for Computational Linguistics: Human Language Technologies, Volume 1 (Long and Short Papers). 2019.

**Questions:**

Please see Weaknesses.

---

> ### Author Response · Authors · 2024-11-23
>
> We thank the reviewer for their valuable suggestions and feedbacks. Below we address the weaknesses and questions in detail.
>
> ### W1-1. Motivation of studying context-awareness
>
> We agree with the reviewer that the goal of instruction-finetuning is to achieve a helpful agent. However, helpfulness should not be treated as an isolated optimization target for instruction finetuning. We believe being aware of the details in the user instruction is an important criteria of a helpful agent. Imagine a scenario where a financial analyst requesting an AI assistant to summarize and analyze a recent news article, missing or hallucinating any single one of the details can be catastrophic. Many real-world applications also require responding to the user requests according to the provided context (i.e. the context-awareness), such as performing question-answering according to information provided in documents. Therefore, the context-awareness of a chat agent is highly relevant to its general helpfulness.
>
> ### W1-2. Context-awareness v.s. general instruction-following
>
> We would like to clarify that with evaluation on **non-contextual** instruction following benchmarks such as MT-Bench, our motivation is to evaluate and make sure the performance does not drop on user requests without a heavy dependency on user specified contexts. And this is validated with our experiment results that after finetuning with the indicator, performance on tasks without heavy context-dependency does not drop and even increases under most settings.
>
>
> ### W2-1. Context-awareness attributed to attention allocation
>
> We would like to first make some clarifications about Figure 3. In Figure 3, we visualize three bars for each model. The second bar contains weight allocation on all tokens, including the chat template. We agree with the reviewer that attention allocated to template tokens is expected. Therefore, we plotted a third bar with only attention allocation on user and response tokens when the chat template is added. Our conclusion is supported by the comparison between the first and third bar – the net attention allocated on user tokens decreased even when  only comparing with the response tokens. To present the results more intuitively, we updated Figure 3 to report the difference between attention scores with and without applying the chat template. The full figure is moved to Appendix B.2.
>
> ### W2-2 Only cherry-picking one attention head
>
> The motivation of selecting a single attention head for visualization is that different attention heads can have very different functionalities. The attention head related to context retrieval can also be very sparse in a pretrained transformer model, which is also validated by a recent work [1]. Therefore, drawing inspiration from the fact that context retrieval heads allocate higher scores to the user context, we reported the attention head with the highest attention score summed over the user context to represent the context awareness of the model. As summing up attention allocations across different layers may not be sound, we only visualize on one layer. To sanity check whether the attention score of the selected attention head is representative of the model, we also compared other retrieval heads from different layers and observed the high agreement in attention score patterns amongst them (see Appendix Fig 4).
>
> ### W-3 Reporting MT-Bench performance for attention-steering
>
> An assumption for the use cases of our proposed methods is that the user can control over the model about whether they want the model to rely more on the user provided context or not. Emphasizing the context is unnecessary for non-contextual instruction following use cases such as MT-Bench and we would expect the users to use the models normally without attention steering and performance will stay the same as the original model.
>
> ### W-4 Not significant improvements in Table 3
>
> We first want to highlight that the overall performance improvement on the four tasks is not marginal, although the performance saturation on easier tasks  may give the impression that the improvement is not as significant. For Llama-3-8B with WizardLM, the performance on SQuAD is even higher than the official Llama-3-8B-Instruct version fine-tuned on closed-source datasets, while for QuAC, Llama-3-8B with WizardLM still achieves improvements from 0.1570 to 0.1710. Similarly, Llama-2-7B with UltraChat achieves a much better performance on QuAC compared to the official instruct version released by Meta, which on SQuAD, the improvement is more obvious from 0.8272 to 0.8508.
>
> We hope our clarifications would address your concerns, and thanks again for your feedbacks.
>
>
> [1] Wu, W., Wang, Y., Xiao, G., Peng, H., & Fu, Y. (2024). Retrieval head mechanistically explains long-context factuality. arXiv preprint arXiv:2404.15574.

---

> > ### Comment · Reviewer_1kxK · 2024-11-23
> > **Comments to Author Response**
> >
> > For W1.1, how do you define context awareness? For example, in your financial analysis case, could it be considered the "truthfulness" properties introduced by OpenAI [1]? If we cannot give a clear definition of this objective, I believe the research problem will be meaningless.
> >
> > For W1.2, the statement "with evaluation on non-contextual instruction following benchmarks such as MT-Bench" is not valid since MT-Bench includes the "Extraction" task [2].
> >
> > For W2.1, your response has not solved my concern. Referring to my main concern, "Finally, instruction-tuned models pay some attention to the chat template, which is intuitive as they are trained on the data with the template; I cannot see any logical connection between the templates and the context awareness (I mean, I cannot prove/derive any connection between these two phenomenons, so they could be correlated, but may not be causally)." One technical point that can naturally lead to this change is that the attention scores are computed after softmax operation, meaning that all outputs are summed up to 1. Thus, since we provide a new component (i.e., chat template) over the initial content (i.e., user prompt) to compute softmax, the attention scores of user prompts will definitely drop.
> >
> > For W2.2, I am satisfied with this discussion on the cheery-pick of a certain attention head.
> >
> > For W3, my concern still holds. Why we can assume a user knows which query relies on a certain capability.
> >
> > For W4, my concern has not been fully resolved. I confirm that in some instances/datasets and training setups, the improvement is significant. However, we cannot constantly see improvement over different settings. It raises my concern on the generalizability of the proposed framework.
> >
> > Since my main concern W1 and W2 still hold, I tend to keep my initial evaluation. In addition, please highlight the modifications in the manuscript.
> >
> > [1] Ouyang, Long, et al. "Training language models to follow instructions with human feedback." Advances in neural information processing systems 35 (2022): 27730-27744.
> > [2] https://github.com/lm-sys/FastChat/blob/main/fastchat/llm_judge/data/mt_bench/question.jsonl

---

> ### Author Response · Authors · 2024-11-23
>
> We thank the reviewer for the follow-up questions. We respond to each question below.
>
> ### W1.1 Definition of context-awareness
>
> As we have mentioned in the introduction section, we define context-awareness as “the capability to extract and understand information from the user-provided context and respond accordingly”. In [1], the paper mainly evaluates the truthfulness of InstructGPT with TruthfulQA (where context is normally not given) and some “closed-domain tasks” (context is given). Our definition of context-awareness is more similar to the capabilities evaluated by the “closed-domain tasks”, where "the output should not contain information that is not present in the input".
>
> ### W1.2 Evaluation on MT-Bench
>
> We admit that there are a few (10 out of 80) extraction queries in MT-Bench. But most of the queries are non-contextual and MT-Bench represents a distribution of general user queries without an emphasis on contextual tasks. In MT-bench evaluations reported in Table 3, we **do not** add the indicator to mimic the general use cases when the user does not add a control to boost the context-awareness. Therefore, it matches our expectation as long as the performance on MT-Bench does not consistently drop (instead the performance increases for most cases).
>
>
> ### W2.1 Visualization of changes of attention allocation
>
> We would like to clarify further that in the third bar of our original Figure 3 we also **renormalized** the attention weight such that **the sum of attention values of user, response and BOS tokens is 1**. We’ve also explained the renormalization in our original caption for Figure 3. In our updated Figure 3, we visualize the relative changes of attention allocation on user and response tokens, between without chat templates and with chat templates and renormalization. We also want to highlight in the updated Figure 3 that when the attention allocation on user token decreases, **attention on response tokens also increases**.
>
>
> ### W3. Why can we assume a user knows which query relies on a certain capability?
>
> The control over context-awareness is an addition to the control over instructions. When the user provides an instruction to the model to “answer the question according to the context”, we can assume that the user should know this query relies more on the user provided context and is able to provide a control with attention steering or indicator. For general use cases where the user does not provide a control to boost the context-awareness, the performance would stay the same for attention steering, and not consistently negatively impacted for finetuning with indicators.
>
> We hope our clarifications would address your concerns, and thanks again for your feedbacks. We’ve also highlighted modifications in our manuscript with blue texts.
>
> [1] Ouyang, Long, et al. "Training language models to follow instructions with human feedback." Advances in neural information processing systems 35 (2022): 27730-27744

---

> ### Comment · Reviewer_1kxK · 2024-11-23
> **Official Comment to Author Response**
>
> For W1.1, what is the difference between your "context-awareness" and "truthfulness" or "hallucination" problems [1]?
>
> For W1.2, I am still waiting for the exact responses to my questions from the initial review: "For example, in Table 3, we can find that a model with better context-awareness performance (i.e., NIH/SQuAD, QuAC, DROP) doesn't necessarily lead to a better instruction-following ability (i.e., MT-Bench). Also, in the NIH example of Figure 1, even though an instruction-tuned LLM cannot provide exactly the same suggestion from the user inputs at the first run, it doesn't necessarily mean that this instruction-tuned LLM lost its ability to retrieve knowledge from the context because it could be the instruction-tuned LLM fail to follow your first instruction that "answer question based on the given paragraph", or even could be the model feels that the target retrieved sentence is not helpful enough. A more comprehensive study on this phenomenon, at least, should provide diverse prompts that emphasize the idea of "answer questions based on the given paragraph" in different levels, and the instruction-tuned LLM constantly fails to retrieve the target sentence."
>
> For W2.1, let me clarify two questions I would like to hear back: (1) What is the formal definition of context-awareness? (2) Why can the distribution of attention weights on user prompts be used to formally quantify context awareness (in your definition)? (3) How your experiment on the distributions of attention weights can conclude to the statement "We identify the performance decline is partially caused by the bias embedded into the chat template to focus less on the the user-provided context." in your abstract?
>
> For W3, I am still waiting for the exact results of attention intervention strategies on MT-Bench.
>
> For W4, my concern has not been fully resolved. I confirm that in some instances/datasets and training setups, the improvement is significant. However, we cannot constantly see improvement over different settings. It raises my concern on the generalizability of the proposed framework.
>
> Since my main concerns W1 and W2 still hold, I tend to keep my initial evaluation.
>
> [1] Huang, Lei, et al. "A survey on hallucination in large language models: Principles, taxonomy, challenges, and open questions." ACM Transactions on Information Systems (2023).

---

### Official Review · Reviewer_bM18 · 2024-10-29

**Soundness:** 3
**Presentation:** 3
**Contribution:** 3
**Rating:** 6
**Confidence:** 4

**Summary:**

This work investigates that the context-awareness ability of instruction fine-tuning models decreases because the attention weight assigned to user prompts decreases when a chat template is added. Based on this observation, they propose an inference-time technique that manually intervening attention scores during response generation. In addition, they also propose a training-time technique that utilizes conditional indicators to further mitigate the loss of context awareness of pretrained language models when instruction-tuning.

**Strengths:**

1. The study on the context-awareness ability of instruction fine-tuned models will decrease is novel. and the observation that the attention weight assigned to user prompts decreases when a chat template is added is interesting and insightful.
2. The proposed  inference-time technique and training-time technique may contribute to the area of instruction fine-tuning.

**Weaknesses:**

The observation that the loss of context awareness after instruction tuning has only been experimented on small LLMs (e.g., llama 7B and 8B), and the models used in the experiment section are also small LLMs equal to or smaller than 8B. Therefore, I do not know whether the loss of context awareness after instruction tuning is a general phenomenon or only occurs on small LLMs, what about the 13B, 70B, and larger models? It is well known that bigger LLMs are more capable than smaller LLMs, whether this phenomenon holds or not is not known, so the study of bigger LLMs is very important for the contribution of this paper to be significant or not.

**Questions:**

Refer to the "Weaknesses".

The observation that the loss of context awareness after instruction tuning has only been experimented on small LLMs, what about the 13B, 70B, and other larger models? This is a limitation of this paper, so:

1. Specific experiments with larger models (e.g., 13B, 70B) that would help validate if the phenomenon generalizes.
2. Explicitly acknowledge this limitation in the paper. Discuss potential implications or hypotheses for how the findings may or may not generalize to larger models.
3. Propose a discussion on how the capabilities of larger language models might interact with or influence the observed loss of context awareness.

---

> ### Author Response · Authors · 2024-11-23
>
> We thank the reviewer for their valuable suggestions and feedbacks. Below we address the weaknesses and questions in detail.
>
> ### W1&W2&Q3. loss of context-awareness for larger models
>
> As requested by the reviewer, **we’ve added two larger models (Llama-2-13B and Gemma-2-27B) in Figure 2 and 3**. The results indicate that larger models still suffer from loss of context awareness after instruction-tuning. For example, the NIH performance gap between with and without applying the chat template on Llama-7b-chat (97% and 99%) persists for the 13b model counterpart (93% and 96%).
> However, when the models are significantly larger, the base performance without chat templates may saturate on the proxy evaluations such as simple NIH and contextual QA tasks, which may result in a less obvious performance drop. This also calls for more difficult evaluation tasks for context-awareness under more practical settings in the future.
>
> ### Q2. Acknowledging the limitations
>
> Thanks for the suggestion about the limitations section. We’ve added discussions about the limitations of model scales considered in the experiments.
>
> We hope our clarifications would address your concerns, and thanks again for your feedbacks.

---

> > ### Comment · Area_Chair_TGML · 2024-11-25
> > **Gentle Reminder from AC**
> >
> > Dear Reviewer bM18,
> >
> > I would like to kindly remind you to take a moment to review the authors' responses for the paper you are reviewing. Nov. 26 is the final day to engage with the authors during the rebuttal phase.
> >
> > Best regards, AC.

---

> > ### Comment · Reviewer_bM18 · 2024-11-25
> >
> > Thanks for your reply, it solves part of my concern, however, model sizes larger than 27b have not been verified and I will keep my score.

---

### Official Review · Reviewer_5xBm · 2024-11-02

**Soundness:** 2
**Presentation:** 2
**Contribution:** 2
**Rating:** 5
**Confidence:** 3

**Summary:**

This paper examines how supervised fine-tuning (SFT) for instruction-following can reduce context awareness in large language models (LLMs). The authors identify a bias from the chat template as a key factor, which shifts focus away from user-provided context. They propose two methods—post-hoc attention steering and conditional instruction fine-tuning with a context-dependency indicator—to counteract this loss. Experiments show these methods effectively restore context awareness without impacting instruction-following ability. The study highlights the importance of benchmarking context awareness post fine-tuning.

**Strengths:**

1. The paper tackles an important topic in LLMs, examining the impact of applying chat templates to inputs during fine-tuning.

2. The paper proposes  both training-free and fine-tuning method to address this issue, evaluated on a range of LLMs and datasets.

**Weaknesses:**

1. The concept of "loss of context-awareness" remains somewhat unclear. Based on the description, it appears related to the contextual reasoning capabilities of LLMs. Section 3.1 illustrates this phenomenon using a chat template in Llama. Does this issue persist across different models and templates? There is a brief discussion in lines 197-199, but no further details are provided.

2. Typo in line 144: “[Optional User template] and [Optional User template] are user and assistant role indicators used.” The second “Optional” should be “Assistant.”

3. Error bars are not reported in the experimental results, such as in Table 2 and Table 3.

4. The experiments are conducted on a limited scope, specifically small-sized Llama models with Q-LoRA. It remains unclear whether the findings would generalize to a broader range of models.

**Questions:**

1. The code has not been provided; how can readers reproduce the results?

2. Could you clarify the concept of "loss of context-awareness" with a more precise definition, illustrative examples, empirical findings, and evaluation metrics?

3. Given the limited time for rebuttal, would it be possible to expand your experiments to a broader scope to strengthen the robustness of the conclusions?

---

> ### Author Response · Authors · 2024-11-23
>
> We thank the reviewer for their valuable suggestions and feedbacks. Below we address the weaknesses and questions in detail.
>
> ### W1. Generalization across different chat templates
>
> We would like to first clarify that in our original Figure 2 we included four models, Llama-2, Llama-3, Mistral-v0.2 and Mistral-v0.3, which contains 3 different chat templates. Therefore, this phenomenon is consistent across different models and chat templates. In our revision, we added evaluations on **three more models in the updated Figure 2 and 3**, including the gemma-2 family and a larger version of llama-2. The results show that the loss of context-awareness, as reflected by the drop of performance on NIH, is consistent among different model families and model sizes.
>
> ### W2. Typo in Figure 1.
>
> Thank you for pointing this out. We’ve updated the caption in revision.
>
> ### W3. Error bars not reported.
>
> As limited by the computational resources, we are not able to fine-tune for multiple times with different random seeds. Therefore, the only randomness is from the inference stage. For evaluation on contextual QA tasks, we use greedy decoding following the default settings of lm-eval library. Therefore, there is no randomness in contextual QA evaluation and we do not report the error bar. For evaluation on NIH, we turned on sampling with the default hyperparameters specified by the pretrained model providers. As limited by computational resources, we are not able to rerun all NIH evaluations multiple times during the rebuttal period. We plan to include error bars for NIH results in future revision.
>
> ### W4. Finetuning limited to QLoRA on 7B/8B models
> Due to limited GPU resources, 7B/8B Llama-2 and Llama-3 are the largest models we are able to finetune with QLoRA on 4 A6000 GPUs. We hope to be able to extend the experiments to larger models if more resources are available in the future.
>
> ### Q1. Code release
>
> We included an anonymous code in our revision, submitted to the supplementary material section. We’ll also release the code publicly upon acceptance.
>
> ### Q2. Definition of context-awareness
>
> In this paper, we define context-awareness as the capability to generate a response according to the information provided in the user input. For example, in the NIH test the answer to the question is given in the user input and the model is explicitly prompted to answer according to the user provided context. Therefore, we can use NIH performance as a proxy for context-awareness of the model.
>
> ### Q3. Extension to more models
>
> Please see our response to W1
>
>
> We hope our clarifications would address your concerns, and thanks again for your feedbacks.

---

> > ### Comment · Area_Chair_TGML · 2024-11-25
> > **Gentle Reminder from AC**
> >
> > Dear Reviewer 5xBm,
> >
> > I would like to kindly remind you to take a moment to review the authors' responses for the paper you are reviewing. Nov. 26 is the final day to engage with the authors during the rebuttal phase.
> >
> > Best regards, AC.

---

### Official Review · Reviewer_yshk · 2024-11-03

**Soundness:** 2
**Presentation:** 2
**Contribution:** 2
**Rating:** 3
**Confidence:** 4

**Summary:**

This paper claims the loss of context-awareness on instruction-tuned LLMs when the chat template is applied to the input prompts. It further proposes two methods to mitigate the loss of context awareness in instruct models: post-hoc attention steering and instruction tuning with special token. Though studying an interesting topic, the paper still has multiple drawbacks.

**Strengths:**

It is interesting to study the loss of context-awareness for instruction-tuned LLMs with chat templates.

**Weaknesses:**

1. I am not convinced about the phenomenon of loss of context awareness for instruction-tuned LLMs with chat templates. Fig. 3 shows attention weight allocation in which a part of the attention is allocated to chat template tokens. However, these format tokens serve to indicate dialogue roles or separate dialogue turns. It is not straightforward to claim the allocation of attention weights to chat template tokens would cause the loss of context awareness.

2. It is not well motivated that the proposed two methods would serve to mitigate the claimed loss of context awareness. I suspect the post-hoc attention steering would be beneficial since the model is further optimized through instruction finetuning. Furthermore, it is unclear why prepending a special token as the indicator to user instruction for instruction tuning would mitigate loss of context awareness.

3. In Table 2, the best performance with alpha=1.0 for multiple scenarios, and the minor difference between alpha=1.0 and alpha=0.9 seem to indicate the post-hoc performance is ineffective to mitigate the claimed loss of context awareness.

4. In Table 3, the improvements with indicator for popular public benchmarks like SQuAD, QuAC, DROP, and MT-Bench seem very small. These results strengthens my concerns on the effectiveness of prepending the special token for mitigating the loss of context awareness.

**Questions:**

None.

---

> ### Author Response · Authors · 2024-11-23
>
> We thank the reviewer for their valuable suggestions and feedbacks. Below we address the weaknesses and questions in detail.
>
>
> ### W1: attention weight allocation on chat template tokens
>
> We would like to clarify some misunderstandings for Figure 3. In our original Figure 3, we visualize three bars for each model. The second bar shows weight allocation on all tokens, including the chat template. We agree with the reviewer that attention allocated to template tokens is expected. Therefore, we draw **a third bar** with attention allocation on **user, response and BOS tokens only** when the chat template is added. We also **renormalize** the attention values such that the attention sum of user, response and BOS tokens is 1, thereby removing the confounding from the template tokens. We can see from the figure that attention allocated on user tokens consistently decreases even when we compare only the user and response tokens. We’ve updated Figure 3 for a clearer presentation for the changes of attention allocation.
>
> ### W2: motivations of the proposed methods
>
> We would like to clarify some misunderstandings in how the proposed methods operate. The post-hoc attention steering is applied on normal instruction-finetuned models and does not require further finetuning. We compare the performance of the same instruction-finetuned model, with and without attention steering. Therefore, the benefits should not come from further optimization.
>
> The second method with the context-dependency indicators requires adding the indicator to selected examples during instruction finetuning. As the model associates the indicator with context-dependent prompts during finetuning, the model will focus more on the user context when  the indicator is presented in the prompt during inference.
> We hope our clarification could better convey the motivation of our methods.
>
> ### W3: marginal improvement from attention steering
>
> Directly intervening the attention values indeed may negatively affect other existing capabilities (also discussed in the third paragraph in Section 4.1). We observe benchmarks that mainly rely on context retrieval benefit more from the attention steering, and less so when the task involves a complex combination of skills (context retrieval being one of the required skills). Specifically, In Table 2 we can see that attention steering (alpha=0.9) achieves better performance on most settings for NIH, which is a more direct evaluation for context-awareness as the answers are exactly contained in the input context. However, other tasks (SQuaD, QuAC, DROP), although require some context awareness, also rely on other skills such as complex reading comprehension and mathematical reasoning. In particular, DROP requires retrieving relevant information from the context AND performing additional calculation or processing on top of the retrieved context to obtain the correct answer. The attention steering potentially hurts the other skills (e.g. basic arithmetics). Therefore, the performance on DROP decreases with attention steering.
>
> ### W4: marginal improvement in Table 3
>
> Among the tasks listed in Table 3, only NIH is designed to directly evaluate context awareness, while other tasks (SQuaD, QuAC, DROP), although require some context awareness, also rely on other skills such as complex reading comprehension and mathematical reasoning. Therefore, performance improvement is more significant on NIH tasks and less on other contextual QA tasks.
> However, we still want to highlight that the overall performance improvement on the four tasks is not marginal. On QuAC, the strongest model considered in the experiments (Llama-3-8B with WizardLM) still achieves improvements from 0.1570 to 0.1710.  However, performance saturation on easier tasks can make the improvement less significant. For Llama-3-8B with WizardLM, the performance on SQuAD is even higher than the official Llama-3-8B-Instruct version fine-tuned on closed-source datasets. The performance saturation on some evaluation tasks also calls for more difficult evaluation tasks for context-awareness.
>
> We hope our clarifications would address your concerns, and thanks again for your feedbacks.

---

> > ### Comment · Area_Chair_TGML · 2024-11-25
> > **Gentle Reminder from AC**
> >
> > Dear Reviewer yshk,
> >
> > I would like to kindly remind you to take a moment to review the authors' responses for the paper you are reviewing. Nov. 26 is the final day to engage with the authors during the rebuttal phase.
> >
> > Best regards, AC.

---

> > ### Comment · Reviewer_yshk · 2024-11-25
> > **After rebuttal**
> >
> > My thanks to the authors for explaining more details. However, I am more concerned about the general motivation of this paper. The paper seems to pay attention to a side effect caused by instruction formats. They call it the loss of context-awareness. However, the paper lacks comprehensive theoretical or experimental proof to show loss of context-awareness due to instruction formats would cause severe performance degradation.
> >
> > The rebuttal did not address my concern above. Thus, I will keep my original rating.

---

> ### Author Response · Authors · 2024-11-30
>
> We thank the reviewer for the post-rebuttal comments. We clarify our motivations of this paper further below.
>
> > However, the paper lacks comprehensive theoretical or experimental proof to show loss of context-awareness due to instruction formats would cause severe performance degradation.
>
> We would like to highlight that in Figure 2, we evaluated 7 off-the-shelf models from 4 model families with different model sizes and chat templates on NIH task, with and without chat templates. The results show that
> 1) NIH performance drops after instruction finetuning, when comparing the green and red bar for each model.
> 2) The performance drop is associated with chat templates, when comparing the blue and red bar for each model.
> These results show that the loss of context-awareness associated with chat templates is consistent among different models.
>
> Note that the performance drop on most models included in our experiments is indeed severe. For example, on Llama-3-8B-Instruct, NIH error averaged among 400 tests increases from 0% to 7.13% when the chat template is added. On Mistral-7B-Instruct-v0.3, NIH error increases from 0.68% to 15.29%.
>
> We hope the clarification can make our motivation clearer.

---

> ### Comment · Reviewer_yshk · 2024-12-01
> **Reply to authors' comments on Nov. 30 2024**
>
> Thanks to the authors for replying to my comments and updating the paper draft. However, from what the revised paper presents, I am still not well motivated to the loss of context-awareness by instruction formats. Here are my thoughts:
>
> 1. The instruction formats are widely used in LLMs nowadays. Given the enhanced capabilities of LLMs to handle longer and longer context window size, the paper presents little evidence to claim instruction format tokens (which typically occupy a small portion of context  window size) would significantly shift the allocation of attention weights and degrade the LLM performance.
>
> 2. The updated paper provides new ablation study of w/ vs. w/o instruction formats on NIH task. However, this ablation study is not convincing to support the paper's claim due to: a) The NIH task is not a major evaluation benchmark widely used for LLM performance evaluation. b) It evaluates only smaller size models (<=13b), and misses important SOTA open-source models like QWen and Llama 3.1. It is reasonable to speculate larger or better trained models with stronger capabilities would be affected less to such degradation.
>
> 3. The new Fig. 2 and Fig. 3 in the revised paper are not persuasive to support the claim. The Fig. 3 shows the percentage change of attention allocation. However, the attention allocation change in the paper is typically within 10% and could be considered small. Fig. 2 only shows the recall error but ignores the precision change. It is not reasonable to analysis only the significancy of recall change (i.e., the 7.13% and 15.29% in the authors' comments) with ignoring the precision.
>
> 4. The proposed methods and their performance gains do not prove their significance in effectively improving the degradation. It indicates either the "loss of context-awareness" is not significant in major LLM benchmarks, or the proposed methods are not effective in alleviating such degradation.
>
> In all, I think the updated paper still contains a lot of drawbacks or questions not fully addressed. I will still keep my original rating.

---

> ### Author Response · Authors · 2024-12-01
>
> We thank the reviewer for the follow-up comments. We respond to each comment and clarify a few misunderstandings in detail below:
>
> > a) The NIH task is not a major evaluation benchmark widely used for LLM performance evaluation
>
> NIH is widely used as a benchmark focusing on contextual retrieval. It's reported by many recent open-sourced models as a direct evaluation for context retrieval capabilities, including qwen2 [1], llama-3 [2], deepseek-v2 [3] and many others. Although there are other benchmarks such as MMLU, HumanEval, MATH, we do not include them in our analysis as the focus of our paper is context-awareness and most queries in these benchmarks do not contain a specific context.
>
> > b) It evaluates only smaller size models (<=13b), and misses important SOTA open-source models like QWen and Llama 3.1.
>
> The reason that we do not evaluate on newer models such as qwen2 and Llama 3.1 is that these newer models are highly likely to have been specifically finetuned on NIH similar tasks. Llama 3.1 report has clearly mentioned this in the report (Please see Section 4.3.4 Long Context in [2]) though qwen-2 doesn't have detailed information about their data preparation. In this case, comparing the instruction finetuned model and its pretrained model is not that informative. Saying that, Llama 3.1 still shows a drop on NIH task in their evaluation. We attach their Table 21 for instruct model evaluation below. Pretrained models are reported to be able to perfectly solve NIH task (please see 3.4.2 Long Context Pre-Training in [2]).
>
> |                    | Llama 3(.1) 8B  | Llama 3(.1) 70B  | Llama 3(.1) 405B   |
> |--------------------|-------------|-------------|--------------|
> | NIH (Multi-Needle) | 98.8 $\pm$ 1.2 | 97.5 $\pm$ 1.7 | 98.1 $\pm$ 1.5 |
>
> > The Fig. 3 shows the percentage change of attention allocation. However, the attention allocation change in the paper is typically within 10% and could be considered small.
>
> We would like to highlight that the trend of increasing attention on response tokens and decreasing attention on user tokens is highly consistent among different models. Besides, the attention shift is not 'small' considering the response prefix contains only 10 tokens.
>
> > Fig. 2 only shows the recall error but ignores the precision change. It is not reasonable to analysis only the significancy of recall change (i.e., the 7.13% and 15.29% in the authors' comments) with ignoring the precision.
>
> We must argue that for NIH task, recall is a more reasonable metric as precision can be largely affected by the model's response style (e.g. some models prefer to add more detailed explanation in their responses) . For example, answer A is 'is to eat a sandwich and sit in Dolores Park on the sunny day.', and answer B is 'is to eat a sandwich and sit in Dolores Park on the sunny day, according to the provided document.'. In this example, the precision of answer A is significantly higher than answer B while they both accurately retrieve the required information. Recall as a metric for NIH test is also used in Llama-3 report [2].
>
> To control the precision while reporting the recall, we trim the model's response up to 50 tokens and report the recall within the first 50 tokens (Appendix A.3). In this way, we avoid reporting a high recall when the precision is extremely low.
>
> > The proposed methods and their performance gains do not prove their significance in effectively improving the degradation. It indicates either the "loss of context-awareness" is not significant in major LLM benchmarks, or the proposed methods are not effective in alleviating such degradation.
>
> Please see our initial responses to W3&W4.
>
> [1] Qwen2 Technical Report
>
> [2] The Llama 3 Herd of Models
>
> [3] DeepSeek-V2: A Strong, Economical, and Efficient Mixture-of-Experts Language Model

---

### Note · Authors · 2024-12-29

I have read and agree with the venue's withdrawal policy on behalf of myself and my co-authors.